# Barriers and enablers for the deployment of large language model-based conversational robots for older adults: A protocol for a systematic review of qualitative studies

**Ravi Shankar** [ID][1]*, **Anjali Bundele**[1,2], **Amartya Mukhopadhyay**[3]

1 Research and Innovation, Medical Affairs, Alexandra Hospital, National University Health System, Singapore, 2 Yong Loo Lin School of Medicine, National University of Singapore, Singapore, 3 Division of Respiratory & Critical Care Medicine, Department of Medicine, National University Health System, Singapore

* Ravi_SHANKAR@nuhs.edu.sg

## Abstract

### Background

Artificial intelligence-powered conversational agents have immense potential to provide social companionship and support for older adults. However, the deployment of large language model (LLM)-based conversational robots for seniors faces various technical, user acceptance, and ethical challenges.

### Objectives

This systematic review aims to synthesize insights from prior qualitative studies to identify key factors that influence the real-world application of LLM-based conversational agents for the aging population. The review will inform the user-centered design of these technologies, policy discussions on their governance, and highlight research gaps.

### Methods and analysis

Eleven electronic databases will be searched for qualitative studies exploring stakeholder perspectives on using AI chatbots and robots to assist seniors. Two reviewers will independently screen studies, extract data, and appraise methodological quality using the JBI checklist. Thematic analysis will be conducted to identify major barriers and enablers, and confidence in review findings will be assessed using the GRADE-CERQual approach. The review will adhere to PRISMA-P and ENTREQ reporting guidelines to ensure transparency.

### Discussion

Understanding and addressing obstacles to implementing LLM-powered conversational agents for older adults is crucial for leveraging this technology to support the well-being of the rapidly aging global population. This systematic review will provide timely insights to guide the responsible development and deployment of AI companions for seniors.

**Data availability statement:** No datasets were generated or analysed during the current study. All relevant data from this study will be made available upon study completion.

**Funding:** The author(s) received no specific funding for this work.

**Competing interests:** The authors have declared that no competing interests exist.

## Trial registration

ClinicalTrials.gov CRD42024601264

## 1. Introduction

### 1.1 Background

The global population is rapidly aging, with the number of individuals over 60 years old projected to reach 2.1 billion by 2050 [1]. This demographic shift presents immense challenges for healthcare and social support systems worldwide. Older adults often experience declines in physical and cognitive functioning, reduced social engagement, and increased need for assistance with activities of daily living (ADLs) [2]. The COVID-19 pandemic has further exacerbated these issues by isolating seniors and disrupting their care [3].

Artificial intelligence (AI) has emerged as a promising tool to support the well-being of older adults. In particular, AI-powered conversational agents, such as chatbots and social robots, can engage seniors in natural interactions to provide companionship, mental stimulation, and task assistance [4]. The recent advent of large language models (LLMs) has dramatically advanced the conversational and reasoning abilities of these AI agents [5]. LLMs are machine learning models trained on vast amounts of natural language data to generate human-like text. By powering chatbots and robots with LLMs, these agents can engage in highly coherent, knowledgeable and context-aware conversations to build rapport with users and cater to their individual needs [6].

Several studies have demonstrated the potential benefits of conversational AI for older adults. For example, Sidner et al. [7] found that a social robot improved loneliness, depression and quality of life in seniors living alone. Vardoulakis et al. [8] showed that an LLM-powered chatbot could provide engaging and comforting interactions for isolated older adults during COVID-19. However, research has also identified significant barriers to the real-world implementation of conversational AI for seniors [9]. These include challenges in designing intuitive interfaces for tech-novice users, concerns about privacy and data security, difficulties in integrating systems with existing care practices, limited digital literacy skills among seniors, and more. For conversational robots powered by massive LLMs, additional obstacles arise regarding the transparency, safety and ethical implications of these advanced black-box models [10].

To realize the potential of LLM-based conversational robots for supporting older adults, it is crucial to understand the complex web of factors that enable or impede their real-world deployment. However, the current evidence base is fragmented across diverse fields like human-robot interaction, natural language processing, gerontology and applied ethics. A systematic consolidation and comparison of findings from existing qualitative studies is lacking. Such a synthesis is needed to comprehensively map out barriers and enablers, explore relationships between them, and derive key learnings to advance research and development in this area for the benefit of the aging population.

### 1.2 Objective

The objective of this paper is to present a protocol for a systematic review of qualitative studies on the barriers and enablers for deploying LLM-based conversational robots to support older adults. The review aims to integrate evidence and insights from multiple disciplines to answer the following research questions:

1. What are the major barriers that impede the implementation of LLM-powered conversational robots for older adults?

2. What are the key enablers that facilitate the deployment and adoption of this technology by seniors and their care networks?

3. How do the identified barriers and enablers interact with each other and what are their relative weights of importance based on qualitative evidence?

4. What are the research gaps and future directions for inquiry on this topic based on the review findings?

By systematically aggregating and interpreting pertinent qualitative findings, this review will provide a timely evidence synthesis to guide the research, design, and deployment of LLM-based conversational robots that can be effectively used by older adults in real-world contexts. The review protocol is reported in accordance with the Preferred Reporting Items for Systematic Review and Meta-Analysis Protocols (PRISMA-P) guidelines [11]. The review itself will follow the Enhancing Transparency in Reporting the Synthesis of Qualitative Research (ENTREQ) statement [12] to ensure methodological rigor.

## 2. Methods

This systematic review commenced in January 2025. The anticipated timeline for completion of key stages is as follows:

- Database searches and record retrieval: March 2025

- Title and abstract screening: April 2025

- Full-text screening and eligibility assessment: May 2025

- Data extraction and quality appraisal: June-July 2025

- Data synthesis and initial draft of findings: August-September 2025

- Manuscript preparation and submission: October 2025

The review is currently in the protocol stage, with database searches to begin following protocol publication. No screening, data extraction, or analysis has been conducted at the time of protocol submission. Any significant deviations from this timeline will be reported in the final review publication.

### 2.1 Eligibility criteria

The eligibility criteria for including studies in this review will be defined using the PICO (Population, Intervention, Comparison, Outcome) framework [13].

**2.1.1 Population.** The population of interest includes:

- Older adults aged 60 years and above

- Formal and informal caregivers of older adults (e.g., family members, home care aides, nurses)

- Other stakeholders involved in the care of older adults or implementation of conversational AI (e.g., healthcare professionals, aging service providers, policymakers, AI/robotics experts)

Studies will be included if they report on the perspectives and experiences of any of these populations regarding the use of conversational agents for older adults. No restrictions will be placed on participants' gender, race, ethnicity, socioeconomic status, or geographic location.

**2.1.2 Intervention.** The intervention of interest is conversational agents designed to provide companionship, assistance, and support to older adults. Conversational agents are defined as AI technologies that use natural language interaction (via text and/or speech) to communicate with users [4]. This includes, but is not limited to:

- Chatbots and voice assistants

- Embodied conversational agents and social robots

- Interactive AI systems powered by large language models (LLMs)

Eligible conversational agents may serve a variety of use cases for supporting older adults, such as:

- Social companionship and reducing loneliness

- Assistance with activities of daily living

- Cognitive training and mental stimulation

- Health monitoring and medication management

- Facilitating connection with family and healthcare providers

The technical architectures and capabilities of the conversational agents may vary, but eligible studies should focus on agents that use natural language as the primary interaction modality. Studies that only discuss non-conversational robots (e.g., rehabilitation or telepresence robots) will be excluded. Studies that examine the perspectives of stakeholders on conceptual, anticipated or hypothetical conversational agents will be included, as long as the agents are described in sufficient detail.

**2.1.3 Comparison.** For this review, the comparison is not a critical element of the eligibility criteria, as the primary aim is to synthesize qualitative evidence on factors that influence the implementation of conversational agents for older adults, regardless of whether studies include explicit comparisons to other interventions or technologies. Where reported, information about comparisons made in eligible studies will be extracted and discussed in the context of understanding barriers and enablers.

**2.1.4 Outcomes.** The outcomes of interest are the perceived barriers and enablers that influence the implementation, adoption and use of conversational agents to support older adults. Barriers are defined as any factors that impede or complicate the deployment of conversational agents in real-world settings, while enablers are factors that facilitate or drive implementation.

Relevant barriers and enablers may operate at multiple levels, including individual (e.g., user acceptance, perceived usability, digital literacy), interpersonal (e.g., family support, caregiver attitudes), organizational (e.g., care provider readiness, training), societal (e.g., data privacy regulations, digital infrastructure), economic (e.g., cost, reimbursement models) and ethical (e.g., transparency, accountability).

The scope of barriers and enablers is kept purposefully broad, as a key aim of the review is to comprehensively map the diversity of factors that shape implementation based on the perceptions and lived experiences of relevant stakeholders. However, barriers and enablers must be meaningfully related to the real-world use of conversational agents for supporting older adults. Factors that are very generic (e.g., general attitudes towards aging) may be excluded if they lack specificity to the review topic.

Studies will be eligible if they report qualitative data relating to perceived barriers and/or enablers. This may include data collected through interviews, focus groups, observation, case

studies, open-ended survey questions, or other qualitative methods. Studies that only report quantitative data (e.g., technology acceptance ratings) will be excluded. Mixed-methods studies will be included if they report qualitative findings relevant to the review.

## 2.2 Information sources and search strategy

A comprehensive literature search will be conducted in the following electronic databases: PubMed, Web of Science, Embase, CINAHL, MEDLINE, The Cochrane Library, PsycINFO, and Scopus. Additional engineering and computer science databases including IEEE Xplore, ACM Digital Library, and Engineering Village will be searched to capture relevant technical studies and pilot implementations that may contain qualitative data about user experiences and implementation challenges. These databases were selected to ensure comprehensive coverage of human-computer interaction and robotics research involving conversational agents for older adults. These databases were selected to capture relevant research from diverse fields, including healthcare, social sciences, human-computer interaction, and AI/robotics. No limits will be placed on publication date, type, or language of publication. For non-English publications, English abstracts will be screened for eligibility, and full texts of potentially eligible studies will be assessed using machine translation tools combined with native speaker verification. This approach ensures comprehensive coverage while maintaining methodological rigor.

Grey literature will be included through targeted searches of relevant conference proceedings, technical reports, theses/dissertations, and organizational websites. Sources will include major gerontechnology and human-robot interaction conferences, research institutions focused on aging and technology, and organizations involved in AI ethics and governance. Including grey literature will help capture emerging findings and reduce publication bias. However, grey literature will be clearly identified and considered separately in the quality assessment and confidence ratings.

The search strategy will use a combination of keywords and controlled vocabulary terms (e.g., MeSH) relating to three key concepts: (1) conversational agents, (2) older adults, and (3) barriers and enablers. The exact search terms will be adapted for each database.

A template search string was developed based on a related systematic review on the perceptions and experiences of peritoneal dialysis patients [14]. This template was modified to fit the specific concepts of interest for the current review. The search string for PubMed is provided below as an example:

("aged"[MeSH Terms] OR "aging"[MeSH Terms] OR "older adult*"[Title/Abstract] OR "elder*"[Title/Abstract] OR "senior*"[Title/Abstract])

AND

("conversational agent*"[Title/Abstract] OR "dialogue system*"[Title/Abstract] OR "conversational AI"[Title/Abstract] OR "social robot*"[Title/Abstract] OR "embodied conversational agent*"[Title/Abstract] OR "chatbot*"[Title/Abstract] OR "voice assistant*"[Title/Abstract] OR "large language model*"[Title/Abstract] OR "LLM"[Title/Abstract] OR "GPT"[Title/Abstract])

AND

("barrier*"[Title/Abstract] OR "obstacle*"[Title/Abstract] OR "challenge*"[Title/Abstract] OR "impediment*"[Title/Abstract] OR "enabler*"[Title/Abstract] OR "facilitat*"[Title/Abstract] OR "driver*"[Title/Abstract] OR "acceptance"[Title/Abstract] OR "adoption"[Title/Abstract] OR "usability"[Title/Abstract] OR "user experience"[Title/Abstract] OR "implementation"[Title/Abstract] OR "deployment"[Title/Abstract] OR "feasibility"[Title/Abstract] OR "desirability"[Title/Abstract] OR "viability"[Title/Abstract])

AND

("qualitative"[Title/Abstract] OR "interview\*"[Title/Abstract] OR "focus group\*"[Title/Abstract] OR "case stud\*"[Title/Abstract] OR "ethnograph\*"[Title/Abstract] OR "grounded theory"[Title/Abstract] OR "phenomenolog\*"[Title/Abstract] OR "content analysis"[Title/Abstract] OR "thematic analysis"[Title/Abstract] OR "discourse analysis"[Title/Abstract] OR "observation\*"[Title/Abstract])

To ensure comprehensive coverage, the reference lists of all included studies and relevant reviews will be hand-searched to identify additional eligible studies. The search process will be reported in detail using a PRISMA flow diagram [15] (Fig 1).

In addition to manual reference list searching, backward citation analysis will be conducted using Google Scholar to identify additional relevant studies that have cited the included papers. This systematic citation tracking will help ensure comprehensive coverage of the literature and identify more recent related work that builds upon the primary studies.

## 2.3  Study selection

All records identified through the database searches will be collated and deduplicated using reference management software. Two independent reviewers will then screen the titles and abstracts of all unique records against the eligibility criteria. Records that clearly do not meet the inclusion criteria will be excluded. Full text articles will be retrieved for all records that appear potentially relevant or where there is uncertainty about eligibility based on title/abstract screening.

The full text articles will be independently assessed by two reviewers using the predefined eligibility criteria. Any disagreements between the reviewers will be resolved through discussion or arbitration by a third reviewer. Reasons for exclusion of full text articles will be recorded and reported. The final list of studies included in the qualitative synthesis will be documented, along with a PRISMA flow diagram depicting the study selection process.

## 2.4  Data extraction

Data will be extracted from each included study using a standardized form developed for this review. The draft data extraction form will be pilot tested on a sample of included studies and refined as needed before full data extraction commences. Two reviewers will independently extract data from each study, with discrepancies resolved through discussion or a third reviewer.

The following information will be extracted from each included study: bibliographic details such as authors, year, title, journal/source, and country; the study aim(s) and research question(s); participant characteristics including age, gender, stakeholder type, and sample size; and details of the study setting and context. If applicable, information about the conversational agent(s) discussed in the study will be included. The study design and methodology will also be documented, encompassing the theoretical framework, sampling strategy, data collection methods, and data analysis approach. Verbatim data extracts related to barriers and enablers, such as participant quotes and author interpretations, will be recorded. Finally, key study findings and conclusions relevant to the review objective will be summarized, along with any reported limitations and conflicts of interest. Further details are provided in Appendix A.

Where studies include both qualitative and quantitative data, only the qualitative data will be extracted. If studies include findings not relevant to the review topic, only the pertinent data will be extracted. Extracted data will be managed using qualitative analysis software to facilitate coding and synthesis.

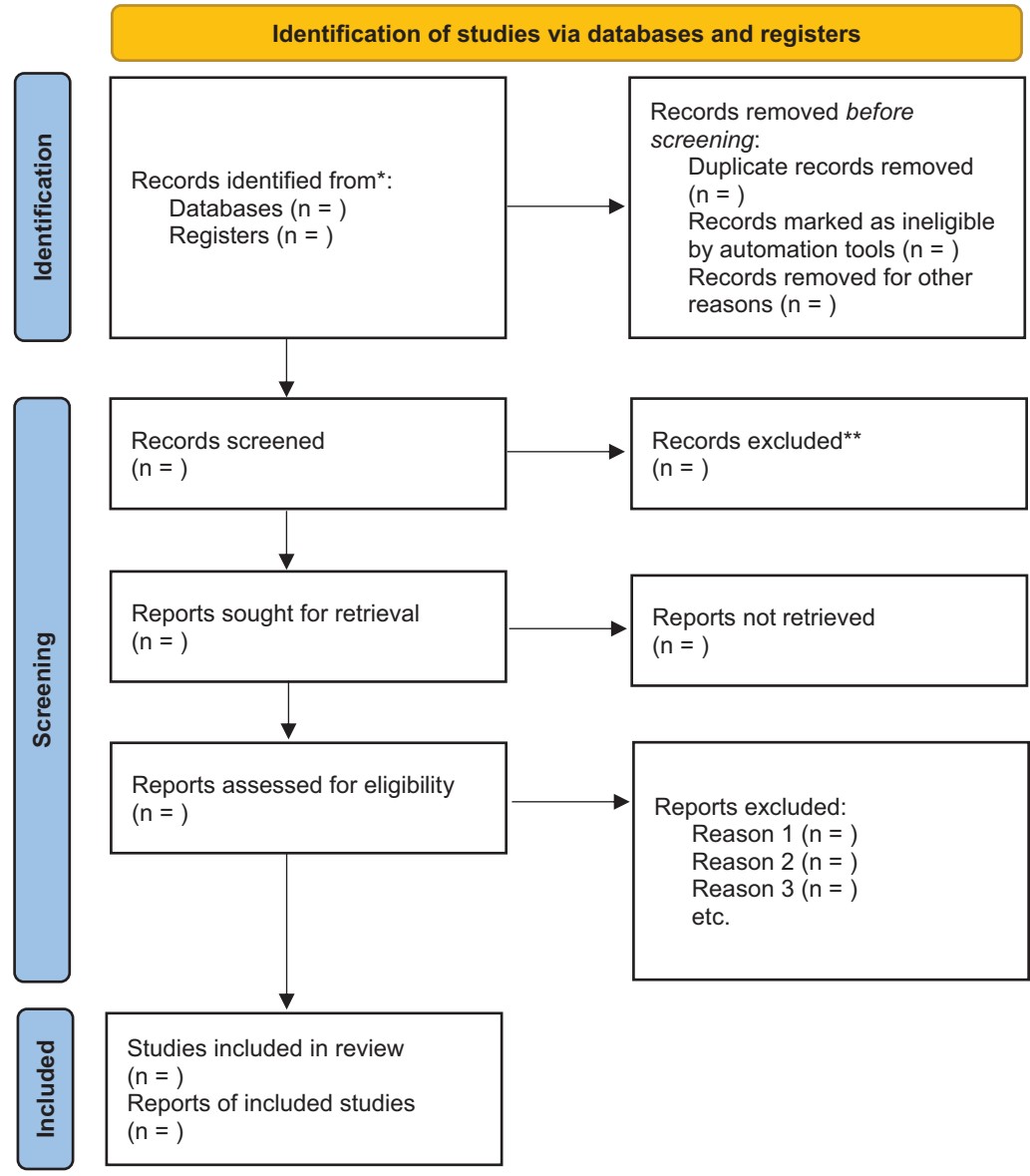

**Fig 1. PRISMA Flow Diagram of Study Selection Process (15).**

## 2.5 Quality appraisal

The methodological quality of included studies will be critically appraised using the Joanna Briggs Institute (JBI) Checklist for Qualitative Research [16]. This tool assesses 10 domains, including congruity between stated philosophical perspective and research methodology, participant voices, researcher reflexivity, and flow of conclusions from analysis.

Two independent reviewers will appraise each study against the JBI checklist, with disagreements resolved through discussion or a third reviewer. Individual checklist items will be rated as "yes", "no", "unclear", or "not applicable". An overall appraisal will be assigned to each study based on whether all, some, or no checklist items are met.

Studies will not be excluded based on quality appraisal results. However, the appraisal findings will be used to inform the confidence placed in each review finding (see section 2.7) and to identify areas for improvement in future research.

## 2.6  Data synthesis

Extracted data will be synthesized using the three-stage thematic synthesis approach developed by Thomas and Harden [17]. Thematic synthesis is an established method for integrating qualitative findings across heterogeneous studies to develop new insights related to a research question. The three stages are:

1. Line-by-line coding of extracted data (participant quotes and author interpretations) to identify specific barriers and enablers.

2. Developing descriptive themes by grouping similar codes together and identifying patterns across studies.

3. Generating analytical themes that go beyond the primary studies by interpreting the descriptive themes in relation to the review question.

The first stage involves inductively coding each line of extracted data as semantic units related to barriers or enablers. Codes will stay close to the original data, using participant terms where possible. Each extract can be assigned multiple codes.

In the second stage, descriptive themes will be developed by looking for similarities and differences between initial codes. Related codes will be grouped into a hierarchical tree structure. Definitions will be developed for each descriptive theme to capture their essence and boundaries.

The final stage will involve interpretively generating analytical themes by 'going beyond' the primary studies. Analytical themes will be developed by considering the descriptive themes in relation to the review questions and theoretical literature. This will result in a synthesis that generates new understandings about the relationships and patterns between different types of barriers and enablers within and across different contexts.

The thematic synthesis will involve a cyclical and iterative process, with constant comparison across data, codes, and themes. Emerging themes will be tested and refined through regular discussion among the review team to enhance rigor. Alternative interpretations will be explored and negative cases sought to challenge the developing synthesis.

Verbatim quotes will be used to illustrate themes and preserve participant voices. Similarities and differences in findings across different populations, settings, and conversational agent characteristics will be explored in the thematic synthesis to enhance the transferability and usefulness of the review findings.

The analytical themes will be synthesized into a final typology or framework of barriers and enablers that influence the implementation and uptake of conversational agents to support older adults. This framework will serve as a novel and practically relevant theory to inform future research, development, and deployment efforts in this field.

## 2.7  Assessing confidence in the synthesis findings

The Grading of Recommendations, Assessment, Development and Evaluations - Confidence in the Evidence from Reviews of Qualitative Research (GRADE-CERQual) approach will be used to assess the confidence that can be placed in each analytical theme based on the primary studies that contribute to it [18]. CERQual provides a transparent and systematic framework for assessing confidence in the evidence, with four components:

1. Methodological limitations of the primary studies

2. Coherence of the review finding

3. Adequacy of data supporting the review finding

4. Relevance of the primary studies to the review question

Methodological limitations will be assessed based on the JBI quality appraisal findings. Coherence will examine how clear and cogent each review finding is. Adequacy will consider the richness and quantity of data supporting each finding. Relevance will assess the applicability of the primary studies to the context specified in the review question.

After assessing each of the four components, an overall confidence judgement will be made for each analytical theme as high, moderate, low, or very low. The CERQual assessments will be conducted by two reviewers, with disagreements resolved through discussion or a third reviewer. The CERQual Evidence Profiles will be presented as supplementary material.

## 2.8 Reflexivity

The author team comprises researchers with backgrounds in gerontology, human-computer interaction, artificial intelligence, and implementation science. Throughout the review process, the authors will engage in regular reflexive discussions to critically examine how their disciplinary orientations, prior knowledge, and underlying assumptions may influence the study design, analysis, and interpretation. Disagreements between reviewers will serve as an opportunity to explore alternative perspectives. The team will keep reflexive notes and share key reflections in the final report to enhance transparency. These practices align with established criteria for maximizing the credibility and usefulness of qualitative evidence syntheses [19].

## 2.9 Ethical approval

Ethical approval is not required for this systematic review protocol as it does not involve human participants or animal subjects.

## 3. Discussion

### 3.1 Strengths and limitations

This protocol has several notable strengths. First, it addresses a timely and consequential question about the real-world implementation of cutting-edge conversational AI to support the well-being of a growing aging population. Second, the eligibility criteria and search strategy are purposefully broad to capture a diverse range of relevant studies across disciplinary and methodological boundaries. The inclusion of multiple stakeholder perspectives and consideration of hypothetical applications alongside real-world examples will enable a more comprehensive understanding of barriers and enablers. Third, the use of established qualitative synthesis methods, quality appraisal tools, and the CERQual approach will enhance the systematic, transparent and rigorous nature of the review.

The inclusion of non-English publications through a systematic translation approach strengthens the review's geographical and cultural representativeness, though we acknowledge that some nuances may be lost in translation despite our verification procedures. The qualitative evidence will be constrained by what was explored and reported in the primary studies, which may not capture all possible barriers and enablers. Publication bias may lead to an under-representation of studies that report predominantly negative or null findings. While our protocol includes non-English publications through machine translation and native

speaker verification, we acknowledge potential challenges in capturing subtle linguistic and cultural nuances through translation. This is particularly important for research from regions like Asia and Europe where there has been significant development and deployment of conversational AI technologies for older adults. To address this, we will work closely with native speaker consultants to ensure accurate interpretation of context-specific implementation factors and cultural perspectives. Additionally, by including publications in local languages, we reduce the risk of publication bias that could occur when relying solely on English publications, as studies with both significant and null findings from non-English speaking countries will be considered, providing a more complete evidence base.

## 3.2  Implications for research and practice

The results of this review will yield a framework of barriers and enablers that can be used to guide research and practice on the implementation of conversational agents to support older adults. By identifying specific factors that influence uptake and mapping the relationships between them, the framework will provide a foundation for developing targeted strategies to overcome key obstacles and leverage facilitators.

The review will inform opportunities for human-centered design of LLM-based conversational agents that are responsive to end-user needs and contextual realities. Insights into older adults' and caregivers' perspectives can guide the co-design of agent personalities, interaction styles, and use case scenarios that enhance usability and acceptance. Organizational leaders and policymakers can use the findings to anticipate and proactively address meso- and macro-level barriers related to digital infrastructure, workforce capacity, data governance, and ethical oversight.

From a research perspective, the review will identify knowledge gaps and key uncertainties that warrant further investigation. The analytic themes may point to novel hypotheses about factors that moderate or mediate the influence of different barriers and enablers. Primary studies could then empirically test these hypotheses using quantitative or mixed methods. The results will also inform the development of new measures to assess contextual determinants of implementation success. Ultimately, the qualitative evidence synthesis will serve as a launch pad for a more strategically coordinated and holistic research agenda to advance the responsible development and deployment of conversational AI for aging societies.

## Author contributions

**Conceptualization:** Ravi Shankar.

**Data curation:** Ravi Shankar, Anjali Bundele.

**Investigation:** Ravi Shankar, Anjali Bundele.

**Methodology:** Ravi Shankar, Amartya Mukhopadhyay.

**Project administration:** Ravi Shankar.

**Resources:** Ravi Shankar, Amartya Mukhopadhyay.

**Software:** Ravi Shankar, Amartya Mukhopadhyay.

**Supervision:** Ravi Shankar, Amartya Mukhopadhyay.

**Validation:** Ravi Shankar, Anjali Bundele, Amartya Mukhopadhyay.

**Visualization:** Ravi Shankar.

**Writing – original draft:** Ravi Shankar.

**Writing – review & editing:** Ravi Shankar, Anjali Bundele, Amartya Mukhopadhyay.

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
