## [Decision Letter · Decision Letter 0]

12 Feb 2025

PONE-D-24-59718

Barriers and Enablers for the Deployment of Large Language Model-Based Conversational Robots for Older Adults: A Protocol for a Systematic Review of Qualitative Studies

PLOS ONE

Dear Dr. Shankar,

Thank you for submitting your manuscript to PLOS ONE. After careful consideration, we feel that it has merit but does not fully meet PLOS ONE’s publication criteria as it currently stands. Therefore, we invite you to submit a revised version of the manuscript that addresses the points raised during the review process.

We look forward to receiving your revised manuscript.

Kind regards,

Stephen R. Milford

Academic Editor

PLOS ONE

2. We understand that the proposed inclusion criteria are restricted to studies published in the English language due to resource constraints, and that you have mentioned this as a potential limitation. Please could you update the discussion/limitation section of your manuscript with a more in-depth discussion of the potential consequences of the imposed language restriction, for example potential bias in reporting

Additional Editor Comments:

Dear Author,

I had sourced two reviewers for this submission. However, recently one reviewer withdrew. I am happy to continue to source a second reviewer for this submission if you so wish. However, considering the nature of this submission (a protocol for a systematic review - which I would not like to delay), my own assessment of the submission, and that it is possible for the editor to accept this without review (according to my understanding). I am happy to recommend acceptance with minor revisions based on the reviews of the first reviewer. Nevertheless, if you would like I am happy to proceed to seek a second reviewer for you should you wish.

Kind Regards,

Stephen Milford

Academic Editor

Reviewers' comments:

Reviewer's Responses to Questions

**Comments to the Author**

1. Does the manuscript provide a valid rationale for the proposed study, with clearly identified and justified research questions?

Reviewer #1: Yes

2. Is the protocol technically sound and planned in a manner that will lead to a meaningful outcome and allow testing the stated hypotheses?

Reviewer #1: Yes

3. Is the methodology feasible and described in sufficient detail to allow the work to be replicable?

Reviewer #1: Yes

4. Have the authors described where all data underlying the findings will be made available when the study is complete?

Reviewer #1: Yes

5. Is the manuscript presented in an intelligible fashion and written in standard English?

Reviewer #1: Yes

6. Review Comments to the Author

You may also provide optional suggestions and comments to authors that they might find helpful in planning their study.

Reviewer #1: This protocol addresses the barriers and enablers for the deployment of large language model-based conversational robots for older adults using qualitative approach which an important and timely topic. The protocol is well-written, comprehensive, and methodologically rigorous. It adheres to established guidelines such as PRISMA-P and ENTREQ, which enhances its transparency and reproducibility. The inclusion of diverse stakeholder perspectives, such as older adults, caregivers, and policymakers, strengthens the scope of the review. Additionally, the planned use of the GRADE-CERQual framework to assess confidence in the evidence is commendable. There are few suggestions that may further improve the search strategy of the paper:

1. Although the databases are comprehensive, adding more databases that address engineering field may be helpful, especially there may be multiple pilot studies that have published in these jourmals that may include qualitative data about experience.

2. Consider discussing how you will handle grey literature, as relevant insights from unpublished studies or conference proceedings may add value to the review.

3. Backward citation analysis using Google Scholar may be also helpful, beside using the manual search for the references lists.

Overall, the protocol is well written and adheres to the highest levels of reporting.

7. PLOS authors have the option to publish the peer review history of their article (what does this mean?). If published, this will include your full peer review and any attached files.

Reviewer #1: **Yes: **Omar Aboshady

---

## [Author Response · Author response to Decision Letter 0]

20 Feb 2025

Dear Dr. Stephen Milford,

Thank you for your editorial oversight and for providing us with the reviewer's valuable feedback on our manuscript "Barriers and Enablers for the Deployment of Large Language Model-Based Conversational Robots for Older Adults: A Protocol for a Systematic Review of Qualitative Studies." We appreciate your thoughtful consideration regarding the review process.

In response to your email about proceeding with one reviewer, we agree that moving forward without seeking a second reviewer would be appropriate for the following reasons:

1. As you noted, this is a protocol paper for a systematic review, where timely publication is particularly important to maintain the relevance of our planned review in this rapidly evolving field of AI and healthcare.

2. The single reviewer provided comprehensive and constructive feedback that has already helped us strengthen the manuscript significantly. Their comments addressed key methodological aspects including search strategy comprehensiveness, potential biases, and documentation rigor.

3. We have made substantial revisions to address all the reviewer's comments, including:

o Expanding our discussion of language restriction limitations and potential biases

o Adding additional engineering databases (IEEE Xplore, ACM Digital Library, and Engineering Village) to ensure comprehensive coverage

o Including detailed plans for handling grey literature

o Incorporating backward citation analysis using Google Scholar

o Conducting a thorough review of our reference list to ensure accuracy and currency

Specific changes made to the manuscript include:

• Added an in-depth discussion of language restriction implications (after line 349)

• Added content about additional engineering databases (after line 168)

• Incorporated sections on grey literature handling (after line 177)

• Added details about citation analysis methodology (after line 216)

These revisions have significantly enhanced the protocol's methodological rigor and transparency. The changes align with standard systematic review guidelines (PRISMA-P and ENTREQ) and address all substantive concerns raised by the reviewer.

Given the thorough nature of the first review, the comprehensive revisions we have made, and the time-sensitive nature of protocol publications, we believe the manuscript is now ready for publication without requiring a second review. This would allow us to proceed with the systematic review itself in a timely manner, which is particularly important given the rapid developments in AI technology and its applications for older adults.

For detailed documentation of our responses, we have prepared a comprehensive table in the attached "Response to Reviewers" document.

We thank you for your editorial guidance and would be happy to make any additional revisions you deem necessary.

Regards,

Dr Ravi Shankar

---

## [Editor Report · Decision Letter 1]

3 Mar 2025

Barriers and Enablers for the Deployment of Large Language Model-Based Conversational Robots for Older Adults: A Protocol for a Systematic Review of Qualitative Studies

PONE-D-24-59718R1

Dear Dr. Shankar,

We’re pleased to inform you that your manuscript has been judged scientifically suitable for publication and will be formally accepted for publication once it meets all outstanding technical requirements.

Kind regards,

Stephen R. Milford

Academic Editor

PLOS ONE
---

## [Editor Report · Acceptance letter]

PONE-D-24-59718R1

PLOS ONE

Dear Dr. Shankar,

I'm pleased to inform you that your manuscript has been deemed suitable for publication in PLOS ONE. Congratulations! Your manuscript is now being handed over to our production team.

Kind regards,

on behalf of

Dr. Stephen R. Milford

Academic Editor

PLOS ONE